

# Improvements to Predictions of the Ionospheric Annual Anomaly by the International Reference Ionosphere Model

Steven Brown [1], Dieter Bilitza [1,2], and Erdal Yiğit [1]

[1]George Mason University, Fairfax, Virginia, USA
[2]Heliospheric Laboratory, NASA Goddard Space Flight Center, Greenbelt, USA Correspondence

**Correspondence:** Steven Brown (sbrown3@masonlive.gmu.edu)

**Abstract.**

The annual anomaly is the ionospheric phenomena in which the globally-averaged electron density is greater in January than it is in July. This anomaly causes the ionospheric solsticial variation—a variation with a periodicity of one year that is in-phase with the January solstice—to be more pronounced over the Northern Hemisphere than the Southern Hemisphere. Predictions of

the magnitude of annual anomaly using the International Reference Ionosphere (IRI) model have been shown to be unreliable so far. The objective of our study is to investigate model prediction of the magnitude of the annual ionospheric anomaly using new ionospheric indices as inputs in the IRI model. These new indices improve predictions ionospheric variations that differ over the two hemispheres. We present a retrospective analysis of the IRI predictions of the ionospheric daytime annual anomaly and solsticial variation using a model-data comparison with observations from over 40 ionosondes for high, moderate, and low

solar cycle conditions. Our results show that there is an overall 33% underestimation of the magnitude of the annual anomaly when the by the IRI. When the new ionospheric indices as used in the IRI, model predictions underestimate the magnitude of the annual anomaly by 6%. This indicates an improvement of the model predictions when using the new indices. We show that the underestimation of the annual anomaly by IRI is related to a similar underestimation of the magnitude of the ionospheric solsticial variation over the Northern Hemisphere. Based on our results, we infer that the underlying processes of the annual

anomaly must vary across each hemisphere.

# 1   Introduction

The annual anomaly is the ionospheric phenomena in which the ionospheric peak electron density, NmF2, is exceedingly greater in January than July. The anomaly also causes the ionospheric climatology to differ over both hemispheres (Rishbeth

and Müller-Wodarg, 2006) The underlying mechanisms of the annual anomaly are not fully understood. The NmF2 is directly proportional to the square of foF2, the maximum frequency of an electromagnetic wave that will be reflected by the ionosphere. Therefore, reliable predictions of these ionospheric parameters are important for various applications which utilize radio wave



propagation through the ionosphere. A reliable ionospheric model that can predict these parameters is a necessary tool that allows users to account for the retarding and refracting effects of the ionosphere on radio wave propagation. The International Reference Ionosphere (IRI) model, as recognized by the International Standardization Organization (ISO) (Bilitza, D et al., 2014; Bilitza et al., 2011), is widely used for the empirical specification of the ionosphere. Through a retrospective model-data
comparison, Rishbeth and Müller-Wodarg (2006) showed that the IRI predictions of the annual anomaly were unreliable. This indicates one aspect of the model predictions which may be improved upon.

Brown et al. (2017) introduced a new ionospheric index, $IG^{NS}$, to be used as the solar cycle input to both of IRI's foF2 models: CCIR-66 (CCIR, 1966) and URSI-88 (Rush et al., 1989) foF2 models. Currently, these models use the 12-month running mean of the official IG index, $IG_{12}$ (also known as the "global sunspot number") (Liu et al., 1983) for solar cycle specification.
The IG index is computed monthly by using CCIR-66 foF2 model predictions at high and low solar activity (where the solar activity is specified by the sunspot number) to convert a set of observed foF2 observations into a set of equivalent sunspot numbers. The median of these equivalent sunspot numbers is selected as the IG index for the given month. Since the IG uses ionospheric observations, it inherently describes solar cycle changes in the ionosphere driven by factors in addition to the solar UV, such as geomagnetic activity and dynamical variations. Through a retrospective model-data comparison using data from
over 50 ionosondes, the IRI foF2 model predictions errors were shown to be significantly reduced when $IG^{NS}$ was used for the solar cycle data input instead of $IG_{12}$. In contrast to $IG_{12}$, $IG^{NS}$ is computed for each hemisphere, instead of the whole globe. It is also not averaged over 12 months. These adjustments improves predictions of temporal and spatial variations the foF2 by the IRI.

We suspect the IRI predictions of the annual anomaly may be improved by utilizing the hemispheric $IG^{NS}$ in place of the
global $IG_{12}$ index. The hemispheric index shows the greatest improvement to IRI model prediction errors over the daytime mid-latitude ionosphere, where observations indicate the ionospheric annual anomaly is more pronounced (Brown et al., 2017; Rishbeth and Müller-Wodarg, 2006). This motivates the premise and primary question for the present study: Does the $IG^{NS}$ index improve IRI predictions of the ionospheric annual anomaly? We suspect that the answer to this question may give further insight into the underlying processes that cause the anomaly.

Observational studies indicate that the foF2 solsticial variation–the ionospheric variation with a periodicity of 1-year and is in phase with the January solstice–differs significantly over the two hemispheres (Torr and Torr, 1973; Richards, 2001; Qian et al., 2013; Zhao et al., 2008). This can be explained if we consider the solsticial variation to be a summation of two independent variations: seasonal and annual variations. Both components have a periodicity of 1 year and are at either a maximum or minimum during a solstice. The annual variation is in-phase with the January solstice over both hemispheres, reaching a
maximum during the January solstice and a minimum during the July solstice. In contrast, the seasonal variation is in-phase with the winter season. During the January solstice, the seasonal variation is at a maximum over the Northern Hemisphere and a minimum over the Southern Hemisphere. The sum of the annual and seasonal variations results in a hemispheric asymmetry in the magnitude of the solsticial variation; the solsticial variation is enhanced over the Northern Hemisphere however, over the Southern Hemisphere (where the seasonal variation is shifted 180 days ahead of the annual variation) the solsticial variation

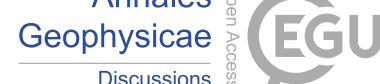



is diminished. These seasonal and annual variations are often associated with two observed ionospheric anomalies: the winter and annual anomalies.

The winter anomaly, which is also referred to as the "seasonal anomaly", is the ionospheric phenomenon in which the electron density is greater in the winter than it is in the summer. This is anomalous because it is expected that the low solar

zenith angle in the winter (relative to that in summer) would instead cause higher electron densities in the summer. Torr and Torr (1973) were among the first researchers to study this phenomenon using monthly observations from over 100 ionosonde stations. They summarized their results in a series of maps detailing the ionospheric solsticial variation (January monthly mean minus July monthly mean) for high, low, and moderate solar cycle conditions. The maps indicate that the winter anomaly is most pronounced over the mid-latitude daytime ionosphere during periods of high solar activity. It is generally accepted that

seasonal changes in the neutral atmospheric composition—the ratio of atomic oxygen to molecular nitrogen—are a predominant underlying cause of the winter anomaly. The enhanced oxygen density in the winter, which is proportional to the electron production rate, is sufficiently large enough to counteract the solar zenith angle effects, thus increasing the electron density in the winter relative to the summer (Rishbeth et al., 2000; Rishbeth, 1998; Yu et al., 2004; Torr et al., 1980; Burns et al., 2014).

The annual anomaly, associated with the annual component of the solsticial variation, is the phenomenon in which the

globally-averaged peak electron density, NmF2, is 30% to 40% higher in January than in July. This is anomalous because the 7% variation in incident solar ionizing radiation, cause by the 3.5% solsticial variation in the Sun–Earth distance, is not sufficient to explain the enhanced electron density. The asymmetry index (AI), as introduced by Rishbeth and Müller-Wodarg (2006), is commonly used to describe the magnitude of the annual anomaly:

$$AI = \frac{A}{M} = \frac{(NmF2_{NS_{Jan}} - NmF2_{NS_{Jul}})}{(NmF2_{NS_{Jan}} + NmF2_{NS_{Jul}})} \tag{1}$$

The AI describes the NmF2 annual variation component (A) relative to its mean variation (M). The AI is computed by using an average of the NmF2 from both Northern and Southern hemisphere which have similar geomagnetic latitudes, in January, $NmF2_{NS_{Jan}}$, and July, $NmF2_{NS_{Jul}}$. A positive AI value indicates that the January values are higher than the July values. Conversely, a negative AI value indicates that the July values are higher than the January values. For example, an AI value of 0.15 indicates that the annual variation is 15% of the mean variation and corresponds to January values of NmF2 that are

approximately 30% higher than the July values.

Numerous scholars have used AI to describe the annual anomaly geophysical conditions, as summarized in Table 1. The AI has been applied to NmF2 retrieved from a ground-based sounding (Rishbeth and Müller-Wodarg, 2006; Mikhailov and Perrone, 2015), satellite-based observations from both topside sounding (Gulyaeva et al., 2014) and radio occultation experiments by the FORMOSAT-3/COSMIC satellite constellation (Zeng et al., 2008; Sai Gowtam and Tulasi Ram, 2017; Momani,

2012). The observed AI values ranged from 0.06 to 0.20, indicating the annual anomaly is real and observable. Observations indicate that the AI tends to increase with solar activity levels, is more pronounced during the day and is more pronounced over the mid-latitude ionosphere (Zhang et al., 2005; Rishbeth and Müller-Wodarg, 2006). The annual anomaly is observable



**Table 1.** Asymmetry Index Applied to Various Ionospheric and Thermospheric Parameters as Reported in Previous Literature

| Author | Year | NmF2 | Thermosphere | Other | Physical Mechanism? |
|---|---|---|---|---|---|
| Mendillo et al. | 2005 | | 0.03 (O/N$_2$, NRLMSIS) 0.06 (O/N$_2$, Observations) | 0.15 (TEC, GPS) 0.035 (Solar EUV) | Compositional Changes in the Neutral Atmosphere |
| Rishbeth and Müller-Wodarg | 2006 | 0.20 (Ionosonde) 0.17 (IRI) -0.036 (CTIP) | 0.02 (CTIP, Tide) 0.13 (O/O$_2$, CTIP) | | Atmospheric tides unlikely; Gravity waves? |
| Zeng et al. | 2008 | 0.14 (COSMIC) 0.11 (GCM) | | 0.035 (F10.7) | Sun-Earth distance; Geomagnetic configuration; |
| Momani | 2012 | | | 0.15 (TEC, GPS) | Neutral wind? |
| Lei et al. | 2013 | 0.05-0.20 (GCM) | 0.15 (LT=12, SAT, rho) 0.13 (GCM, rho) | | Neutral temperature variation driven by Sun-Earth distance |
| Gulyaeva et al. | 2014 | 0.1-14 (SAT, Sea) 0.07-0.06 (ISIS, LSA, Land) | | 0.16 (GEC,GPS) | |
| Zhang et al. | 2014 | 0.18 (660-710km) | -0.06 (500km, NRLMSIS) 0.10 (O, NRLMSIS) | | Variations in neutral composition |
| Ma et al. | 2015 | | 0.09-0.23 (SAT, rho) 0.11-0.22 (NRLMSIS) | | Neutral temperature driven by Sun-Earth Sun-Earth distance |
| Mikhailov and Perrone | 2015 | | 0.29 (N$_2$,NRLMSIS) 0.13 (O, NRLMSIS) | | Enhanced O$_2$ photo-dissociation driven by Sun-Earth distance |
| Sai Gowtam and Tulasi Ram | 2017 | 0.10-0.20 (COSMIC) | | | |
| Dang et al. | 2017 | 0.14 (TIE-GCM) | | | Compositional Changes driven by Sun-Earth distance |

in additional ionospheric parameters such as the total electron content (TEC) and global electron content (GEC) with reported AI values of 0.15 (Mendillo et al., 2005; Momani, 2012; Gulyaeva et al., 2014).

To explain the underlying mechanisms of the annual anomaly, the AI has also been applied to additional solar and thermospheric parameters. Computing the AI for the F10.7 index and the solar irradiance (as observed by TIMED/SEE), yields

an AI value of 0.035. This indicated that the solsticial variation of the incident solar irradiance was not sufficient to explain the annual anomaly (Mendillo et al., 2002; Zeng et al., 2008). Zeng et al. (2008) also used a series of controlled simulations with the thermosphere-ionosphere electrodynamics global circulation model (TIEGCM) to study the annual anomaly. Their results indicated that the offset of the geomagnetic center from the geographic center and the Sun–Earth distance variation significantly contribute to the magnitude of the annual anomaly. When the AI was applied to thermospheric parameters, such

as the atomic oxygen and molecular nitrogen number densities and the total mass density (rho), as predicted by NRLMSISE-00 (Picone et al., 2002), the AI is reported to vary between 0.1 and 0.3. This indicated that variations in the neutral atmospheric composition may play a role in the annual anomaly (Ma et al., 2015; Mikhailov and Perrone, 2015). Mikhailov and Perrone (2015) used empirical predictions of the neutral composition to show that the annual variation in atomic oxygen near the F2 peak accounts for a majority of the annual anomaly. They suggested that the annual variation of the atomic oxygen was a result

of the increased photo-dissociation of molecular oxygen below the F2 layer of the ionosphere. However, this mechanism was challenged in the work by Lei et al. (2016). Through a series of controlled simulations using the global mean model (GMM), which includes the O$_2$ photo-dissociation mechanism, Lei et al. (2016) showed that the mechanism proposed by Mikhailov and Perrone (2015) does not significantly contribute to the annual anomaly. Dang et al. (2017) also used controlled simulations by the TIEGCM to indicate that compositional changes driven by the Sun-Earth distance variations significantly contributed

to the magnitude of the annual anomaly. Despite these efforts, the underlying mechanisms of the anomaly remains a topic of continued research.



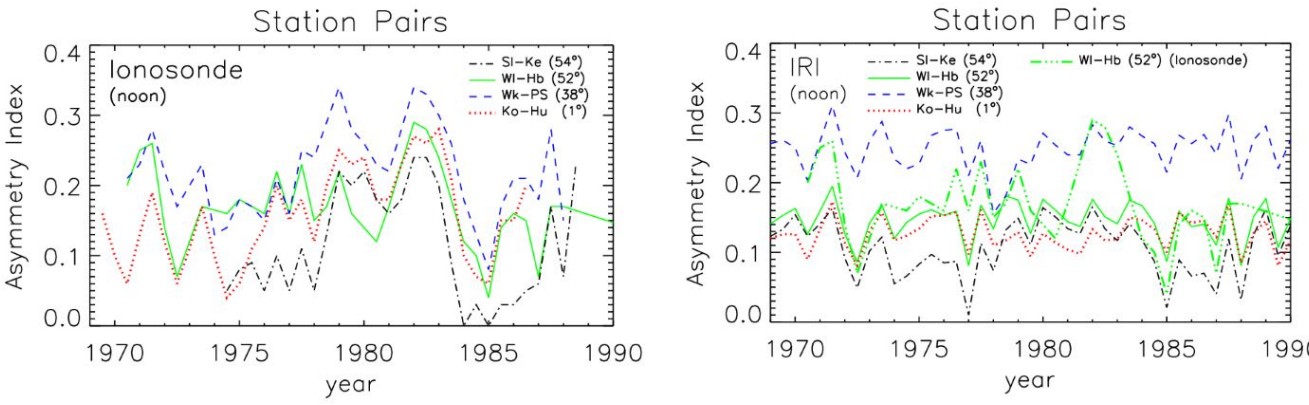

**Figure 1.** Taken from Figures 3 (left) and 4 (right) of Rishbeth and Müller-Wodarg (2006).Variations of the annual anomaly index AI using NmF2 from four Northern/Southern hemisphere station pairs over two solar cycles with ionosonde observations (Left) and IRI predictions (Right).

Few studies of the annual anomaly have involved utilized the IRI model. Rishbeth and Müller-Wodarg (2006) did a model-data comparison of the magnitude of the annual anomaly using IRI predictions and ionosonde observations. Figure 1, which is based on Figures 3 and 4 from Rishbeth and Müller-Wodarg (2006), depicts an example from their model-data comparison where the AI is computed continuously for two solar cycles. The IRI predictions of the AI did not agree well with the ionosonde

results. In general, the IRI underestimated of the magnitude of the annual anomaly by the IRI and also did not reproduced the observed solar cycle behavior of AI. The authors did not comment further on why. Mikhailov and Perrone (2015) investigated the annual anomaly during the 2008–2009 deep solar minimum period using ionosonde observations, and they used predictions by the URSI-88 foF2 model in lieu of the missing data. However, they did not comment on the reliability of the model predictions of the anomaly.

The objective of our study is to investigate IRI predictions of the ionospheric annual anomaly (and solsticial variation) using the hemispheric IG index, $IG^{NS}$, as the solar cycle input. In this paper, we present a retrospective model-data comparison of the annual anomaly using the different IG indices. We suspect the results will further highlight the utility of the $IG^{NS}$ index with regards to IRI model predictions, as well as help to infer the underlying mechanisms of the annual anomaly. This paper is organized as follows. In Section 2, we present the methodology and data sources that were used for this work. In Sections 3.1

and 3.2, we present the model-data comparison results pertaining to the annual anomaly and solsticial variation, respectively. Section 4 presents a discussion of the results and their implications. We conclude with Section 5, which summarizes the present work.





## 2    Methods and Data Sources

In this study, we present retrospective model-data comparison of IRI predictions of the ionospheric annual anomaly and solstical variation using the different IG indices (IG, $IG_{12}$ and $IG^{NS}$) as solar cycle input. For this study, empirical predictions of the foF2 are specified by the URSI-88 foF2 model, the overall recommended IRI foF2 model option (Bilitza, D et al., 2014).

$IG^{NS}$ is computed using the methodology described in Brown et al. (2017). For this study, $IG^{NS}$ was computed using the URSI-88 foF2 model as recommended by Brown et al. (2017). The official IG and $IG_{12}$ indexes were retrieved from the UK Solar System data center (https://www.ukssdc.ac.uk/wdcc1)

### Map of Ionsonde Stations Used for Study

**Figure 2.** Global map of the ionosonde stations used in this solsticial variation comparison.

Figure 2 is a map of the 45 ionosonde stations from which foF2 observations were obtained. Ionosonde data records were retrieved from both the SPIDR (http://spidr.ionosonde.net/spidr/; this service was unfortunately recently discontinued) and

DIDBASE (http://ulcar.uml.edu/DID) data depositories. DIDBASE is maintained by the Global Ionospheric Radio Observatory (GIRO) (Reinisch and Galkin, 2011). These stations were selected because they have at least 8 years of consistent, reliable foF2 measurements, extending from 1970 to 2014, and fall within a geographic latitude range of 30° to 60°. Figure 2 indicates a bias of observations towards the Northern hemisphere therefore two exceptions were made to include two low latitude stations in the North Australian continent: Townsville (GLAT = 21°S) and Learmonth (GLAT = 22°S). This was done to increase

the number of stations representative of the Southern Hemisphere. To minimize any autoscaling errors, only foF2 data that falls between 2 and 20 MHz was used. We further inspected the data and removed clear outliers or sections of data where the ionogram autoscaling may have been in error due to spread F and sporadic E-layer events or numerical floating-point errors. To consider quiet geomagnetic conditions we only used observations corresponds with days in which the an AP index was less

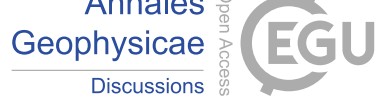

than 30. Daytime January and July monthly medians were computed from the foF2 data which fell between the 10LT–14LT local time bin.

**Table 2.** The Station pairs used to compute the AI. The IGRF model was used to compute the approximate magnetic latitudes at a height of 300km.

| Station Pair | | | Geographic Coordinates | | |
|---|---|---|---|---|---|
| North | South | Abbreviation | LAT | LON | MLAT (N; S; Mean) |
| Wallops | Hobart | Wp-Hb | 38N; 43S | 75W; 147E | 50; 54; 52 |
| Poitiers | Christchurch | Po-Ch | 47N; 44S | 0E; 173E | 43; 48; 46 |
| Eglin | Norfolk | Eg-No | 30N; 29S | 78W; 168E | 42; 36; 39 |
| Wakkanai | Port Stanley | Wa-Po | 45N; 52S | 142E; 58W | 38; 37; 38 |
| Akita | Townsville | Ak-To | 40N; 19S | 140E; 147E | 33; 28; 30 |
| Kodaikanal | Huancayo | Ko-Hu | 10N; 12S | 78E; 75W | 1; 1; 1 |
| Boulder | Hobart | Bo-Ho | 40N; 43S | 255E; 147E | 48; 50; 49 |
| Athens | Canberra | At-Can | 38N; 35S | 24E; 149E | 36; 42; 39 |
| Gibilmanna | Grahamstown | Gi-Gr | 38N; 34S | 14E; 27 | 38; 34; 36 |
| Gibilmanna | Hermanus | Gi-He | 38N; 34S | 14E; 19E | 38; 34; 36 |
| Athens | Grahamstown | At-Gr | 38N; 33S | 24E; 27E | 36; 34; 35 |
| Millstone Hill | Hobart | Mi-Ho | 42N; 43S | 255E; 19E | 51; 50; 50 |

We used the AI index to quantitatively describe the magnitude of the annual anomaly. The station pairs for calculating the AI were selected by pairing Northern and Southern hemispheric stations with similar geomagnetic latitudes, as listed in Table 5    2. To get the station parings, we use the International Geomagnetic Reference Field (IGRF) model to compute the approximate magnetic latitudes at a height of 300 km.

The AI is computed using the NmF2. To compute the NmF2, we use the following formula:

$$NmF2/m^{-3} = 1.24 * 10^{10}(foF2/MHz)^2 \qquad (2)$$

Here, the NmF2 is in units of m$^{-3}$ and the foF2 is in units of MHz. Similar to Torr and Torr (1973), we describe the foF2 10    solsticial variation as the difference between the foF2 values during the January and July solstices:

$$S_{foF2} = foF2_{Jan} - foF2_{Jul} \qquad (3)$$

In Equation (3), the solstice difference, S$_{foF2}$ is computed using foF2$_{Jan}$ and foF2$_{July}$ the monthly median foF2 in January and July, respectively. The S$_{foF2}$ and the AI are computed for various levels of solar activity, where we use the index IG$^{NS}$ as a solar proxy. Solar indices such as F10.7 and the solar sunspot number depart from linearity with the foF2 high solar activity 15    periods. At a certain threshold value, the solar indices continue to increase while the foF2 does not; this is known as the "saturation" effect (Perrone and Franceschi, 1998). During the 2007–2009 deep solar minimum, the solar EUV (and hence, the foF2) descended to anomalously-low values, up to 15% lower than they were during the previous solar minimum. However, the F10.7 values were only 5% lower than previous than they were during the previous solar minimum while there were extended



periods of zero sunspot number. These cases demonstrate the difficulty of distinguishing between high and low solar activity periods when using the solar indices (Chen et al., 2011). The ionospheric $IG^{NS}$ is (by design) linear with foF2 for all activity conditions. Using the 12-month running mean of the $IG^{NS}$, $IG^{NS}{}_{12}$, we easily distinguish periods of solar maximum and solar minimum from one another.

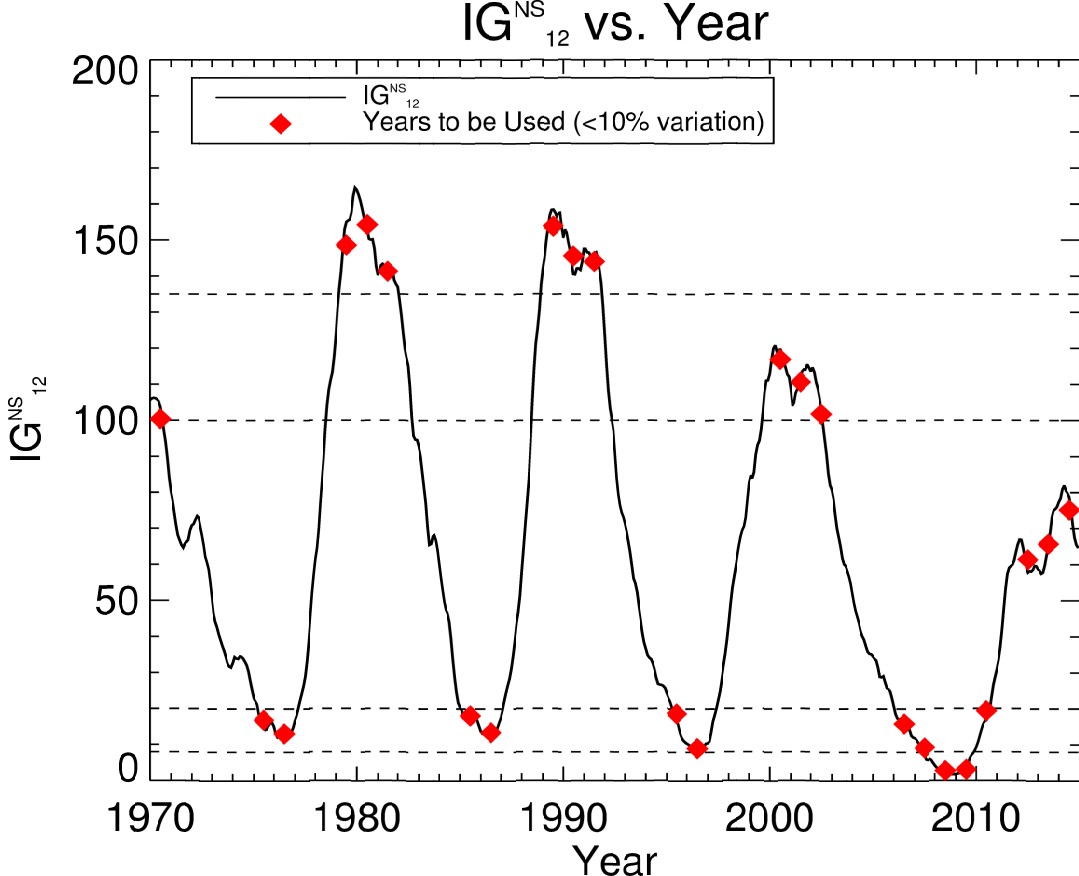

**Figure 3.** The 12-month running mean of $IG^{NS}$, $IG^{NS}{}_{12}$, as a function of time. The horizontal dotted lines indicate "regions" for which the solstices differences are calculated: deep low ($IG^{NS}{}_{12} < 8$), low ($8 < IG^{NS}{}_{12} < 20$), low moderate ($50 < IG^{NS}{}_{12} < 80$), high moderate ($100 < IG^{NS}{}_{12} < 135$) and high ($135 < IG^{NS}{}_{12}$). Only years in which $IG^{NS}{}_{12}$ changes by less than 10% are considered (red diamonds).

5      Figure 3 shows how the years from 1970-2014 are separated by solar activity levels using $IG^{NS}{}_{12}$. We separated the solar maximum periods into high (solar cycles 21 and 22), high moderate (solar cycle 20 and 23), and low moderate (solar cycles 24) solar activity levels. The low solar activity periods are separated into low (solar cycles 21, 22, and 23) and deep low periods (solar cycle 24). We only consider years in which $IG^{NS}{}_{12}$ changed by less than 10% in order to minimize changes in





the ionospheric conditions that may not be associated with variations in the incident solar radiance caused by the Sun–Earth distance variation. A summary of the solar activity levels and selected years is presented in Table 3.

**Table 3.** The solar activity levels, and corresponding years, as defined by the $IG^{NS}_{12}$ index range in value.

| Level | $IG^{NS}_{12}$ | Years |
|---|---|---|
| Deep Low | $IG^{NS}_{12}<8$ | 2008, 2009 |
| Low | $8<IG^{NS}_{12}<20$ | 1975, 1976, 1985, 1986, 1995, 1996, 2006, 2007 |
| Low Moderate | $50<IG^{NS}_{12}<80$ | 2012, 2013, 2014 |
| High Moderate | $135<IG^{NS}_{12}<100$ | 1970, 1982, 1988, 2000, 2001, 2002 |
| High | $IG^{NS}_{12}>135$ | 1979, 1980, 1981, 1989, 1990, 1991 |

## 3 Results

### 3.1 AI at Different Solar Cycle Levels

**Table 4.** Summary of AI calculated with ionosonde data and with IRI using different indices for lower solar activity. '—' indicates ionosonde data were not available for the corresponding solar cycle level.

| Station-Pair | Deep Low | | | | Low | | | |
|---|---|---|---|---|---|---|---|---|
| | Iono | IRI($IG_{12}$) | IRI(IG) | IRI($IG^{NS}$) | Iono | IRI($IG_{12}$) | IRI(IG) | IRI($IG^{NS}$) |
| Wp-Hb | — | 0.10 | 0.09 | 0.12 | 0.15 | 0.12 | 0.05 | 0.15 |
| Po-Ch | 0.05 | 0.03 | 0.01 | 0.05 | 0.15 | 0.05 | -0.03 | 0.08 |
| Eg-No | 0.24 | 0.22 | 0.21 | 0.25 | 0.23 | 0.23 | 0.16 | 0.25 |
| Wa-Po | — | 0.24 | 0.22 | 0.26 | 0.21 | 0.26 | 0.19 | 0.28 |
| Ak-To | — | 0.37 | 0.37 | 0.39 | 0.33 | 0.36 | 0.30 | 0.38 |
| Bo-Ho | 0.14 | 0.10 | 0.08 | 0.12 | 0.15 | 0.12 | 0.04 | 0.14 |
| At-Ca | 0.09 | 0.02 | 0.01 | 0.05 | 0.08 | 0.04 | -0.04 | 0.06 |
| Gi-Gr | 0.10 | 0.08 | 0.07 | 0.11 | 0.09 | 0.09 | 0.01 | 0.11 |
| Gi-He | 0.09 | 0.06 | 0.05 | 0.09 | — | 0.07 | 0.00 | 0.10 |
| At-Gr | 0.12 | 0.07 | 0.05 | 0.09 | 0.16 | 0.07 | -0.01 | 0.10 |
| At-He | 0.11 | 0.04 | 0.03 | 0.08 | — | 0.06 | -0.02 | 0.08 |
| Mi-Ho | 0.14 | 0.10 | 0.09 | 0.12 | 0.11 | 0.12 | 0.05 | 0.15 |
| Average: | 0.13 | 0.09 | 0.08 | 0.12 | 0.19 | 0.16 | 0.09 | 0.19 |
| Average*: | — | 0.14 | 0.13 | 0.16 | — | 0.15 | 0.08 | 0.17 |

Tables 4, 5 and 6 present a summary of the AIs computed for the various solar activity levels. We present results using the ionosonde observations (labeled "iono" ) and predicted by the IRI model using $IG_{12}$, IG, or $IG^{NS}$ as the solar cycle inputs. Two averages are presented at the bottom of the tables: (1) an average that only includes the stations with available data and (2) an average of all the predicted AI values from the IRI model that correspond with the solar cycle level (indicated by a "*"). The average observed AI varies from 0.13 to 0.19 but does not indicate a solar cycle variation. On average, the IRI-predicted AI is



33% lower than the observed AI for all of the presented solar cycle levels when using $IG_{12}$ as solar cycle input. When using the standard monthly IG, the IRI-predicted AI was 37% lower than the observations. When using the $IG^{NS}$, the IRI predicted AI were on average 6% lower than the observations. This indicates that the IRI model, which uses the $IG_{12}$ for solar cycle data input by default, currently underestimates the magnitude of the annual anomaly, and that this can be improved by using the

$IG^{NS}$. What follows is a detailed discussion of each of the solar cycle level results.

Table 4 presents the data for the deep low and low solar activity levels. For the deep low solar activity level, the AI values computed with the ionosonde station observations range from 0.05 to 0.24 and average to 0.13. The predictions by the IRI using $IG_{12}$ vary from 0.03 to 0.21 and average to 0.09 while with IG, the overall average is 0.08. The individual station AI values from the IRI are, at most, 0.08 less than those from the observations. This indicates an underestimation of the AI by the

IRI model when using either $IG_{12}$ or IG as the solar cycle input. When using $IG^{NS}$, the average AI is 0.12, which most closely agrees with the observations. At individual stations, the AI values from the IRI model using $IG^{NS}$ differ from the observed values by 0.03 at most. This indicates that using $IG^{NS}$ yields an AI that is in best agreement with the observed AI at the deep solar solar activity level.

The AI values are underestimated by the IRI model under low solar activity conditions when the $IG_{12}$ or IG indices are used.

The AI values from the ionosonde observations range from 0.06 to 0.41, and they average to 0.19. Overall, the predictions of the IRI model range from -0.04 to 0.32. They average to 0.16 when using $IG_{12}$, 0.09 with IG, and 0.19 with $IG^{NS}$. Using $IG^{NS}$ gives the most accurate AI prediction of AI at the low solar cycle level. In this example, using IG in place of $IG_{12}$ during a low solar activity period worsened the AI predictions; the predicted AI are either negative or zero. This is surprising because the observations indicate the AI is always greater than 0.08. The monthly $IG^{NS}$ shows AI values that consistently agree with the

ionosonde observations, better than when using $IG_{12}$.

**Table 5.** Summary of AI calculated with ionosonde data and with IRI using different indices for moderate solar activity. '—' indicates ionosonde data were not available for the corresponding solar cycle level.

| Station-Pair | Low Moderate | | | | High Moderate | | | |
|---|---|---|---|---|---|---|---|---|
| | Iono | IRI($IG_{12}$) | IRI(IG) | IRI($IG^{NS}$) | Iono | IRI($IG_{12}$) | IRI(IG) | IRI($IG^{NS}$) |
| Wp-Hb | 0.08 | 0.11 | 0.12 | 0.12 | 0.20 | 0.11 | 0.11 | 0.17 |
| Po-Ch | — | 0.06 | 0.08 | 0.07 | — | 0.07 | 0.07 | 0.13 |
| Eg-No | — | 0.17 | 0.19 | 0.21 | — | 0.15 | 0.15 | 0.22 |
| Wa-Po | — | 0.23 | 0.25 | 0.26 | — | 0.23 | 0.23 | 0.28 |
| Ak-To | — | 0.28 | 0.29 | 0.32 | — | 0.23 | 0.23 | 0.30 |
| Ko-Hu | — | 0.11 | 0.12 | 0.16 | — | 0.10 | 0.10 | 0.16 |
| Bo-Ho | 0.13 | 0.11 | 0.12 | 0.12 | 0.20 | 0.11 | 0.11 | 0.16 |
| At-Ca | 0.05 | 0.02 | 0.04 | 0.05 | — | 0.01 | 0.01 | 0.08 |
| Gi-Gr | 0.11 | 0.06 | 0.07 | 0.09 | — | 0.04 | 0.04 | 0.10 |
| Gi-He | 0.15 | 0.05 | 0.07 | 0.08 | — | 0.04 | 0.04 | 0.10 |
| At-Gr | 0.13 | 0.05 | 0.06 | 0.08 | 0.13 | 0.03 | 0.03 | 0.09 |
| At-He | 0.15 | 0.04 | 0.06 | 0.07 | — | 0.03 | 0.03 | 0.10 |
| Mi-Ho | — | 0.12 | 0.14 | 0.13 | 0.20 | 0.13 | 0.13 | 0.19 |
| Average: | 0.11 | 0.06 | 0.08 | 0.09 | 0.18 | 0.09 | 0.10 | 0.15 |
| Average*: | — | 0.12 | 0.14 | 0.15 | — | 0.11 | 0.11 | 0.17 |





**Table 6.** Summary of AI calculated with ionosonde data and with IRI using different indices for high solar activity. '—' indicates ionosonde data were not available for the corresponding solar cycle level.

| Station-Pair | Iono | IRI(IG$_{12}$) | High IRI(IG) | IRI(IG$^{NS}$) |
|---|---|---|---|---|
| Wp-Hb | 0.13 | 0.12 | 0.13 | 0.17 |
| Po-Ch | 0.10 | 0.07 | 0.08 | 0.14 |
| Eg-No | 0.22 | 0.15 | 0.16 | 0.21 |
| Wa-Po | 0.25 | 0.23 | 0.23 | 0.28 |
| Ak-To | 0.22 | 0.22 | 0.23 | 0.28 |
| Ko-Hu | 0.20 | 0.10 | 0.10 | 0.14 |
| Bo-Ho | 0.14 | 0.11 | 0.12 | 0.18 |
| At-Ca | — | 0.01 | 0.02 | 0.07 |
| Gi-Gr | — | 0.03 | 0.03 | 0.09 |
| Gi-He | — | 0.03 | 0.04 | 0.09 |
| At-Gr | — | 0.02 | 0.03 | 0.09 |
| At-He | — | 0.03 | 0.03 | 0.09 |
| Mi-Ho | — | 0.14 | 0.15 | 0.21 |
| Average: | 0.19 | 0.15 | 0.15 | 0.21 |
| Average*: | — | 0.10 | 0.11 | 0.16 |

Table 5 indicates, the IRI predictions of the AI under moderate solar activity conditions underestimate the observed AI when IG or IG$_{12}$ is used for solar cycle data input. The ionosonde AI averages to 0.11 and ranges from 0.08 to 0.15. The IRI predictions that were made using IG$_{12}$ tend to underestimate the AI, showing an average value of 0.06 that varies from 0.06 to 0.29. Using IG yields an AI average of 0.08. Again, using IG$^{NS}$ shows the best agreement of between the predictions and the ionosonde observations averaging to 0.09. There is very limited data for high moderate solar cycle conditions however, results demonstrate an overall underestimated annual anomaly by IRI. Table 6 shows that the IRI predictions of the AI underestimates the observed AI when IG or IG$_{12}$ is used as the solar cycle input under high solar activity level conditions. We point out that the underestimation by the IRI model is the greatest under high solar activity conditions. On average, the ionosonde predictions show an average AI of 0.19 while the IRI predictions show an average AI of 0.15, with both IG$_{12}$ and IG. Using IG$^{NS}$ leads to slight overestimates compared to the observations, because the average is 0.21. However, this is still an improvement over the use of IG$_{12}$ and IG.





### 3.1.1   Solar Cycle Variation of AI

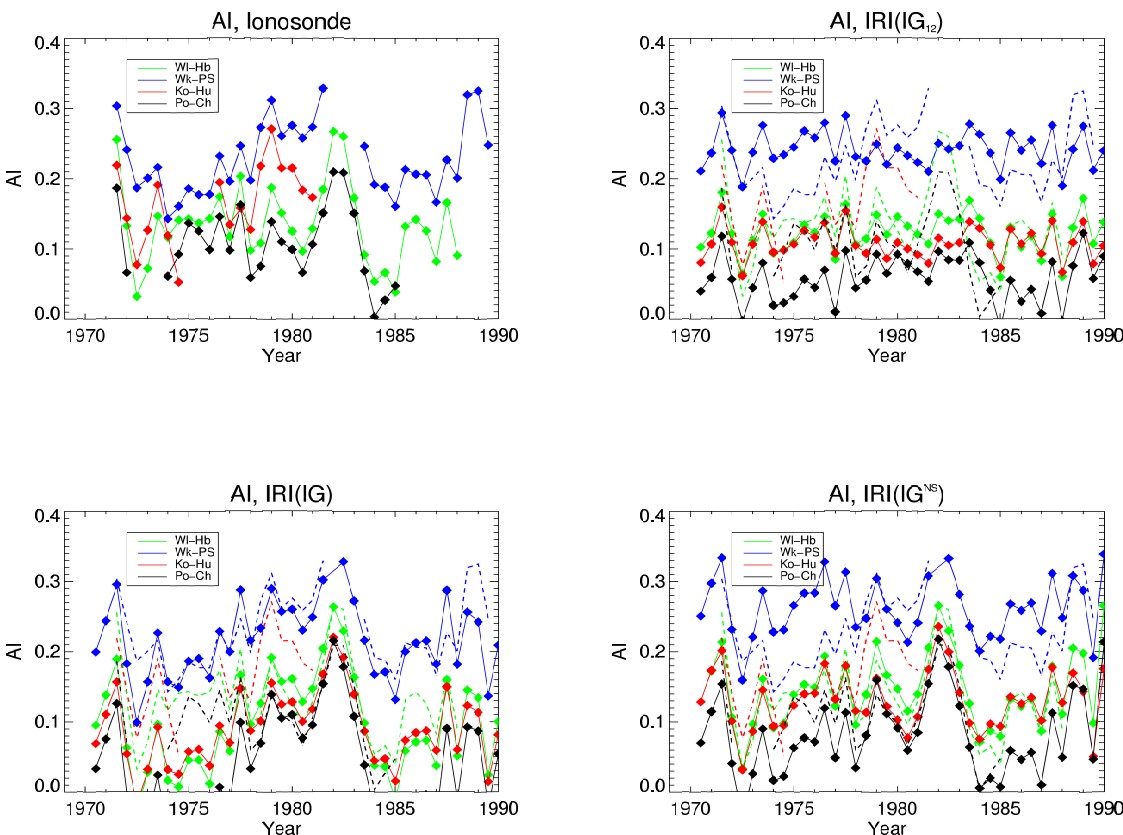

**Figure 4.** The Solar cycle variation of the AI as a function of time using the ionosonde observations (top left) and the IRI model where the solar inputs are varied between $IG_{12}$ (top right), IG (lower left) and $IG^{NS}$ (lower right). We also re-plot the ionosonde-derived AI on each chart as a dotted line for the ease of visual inspection.

Figure 4 presents the solar cycle variation of the AI computed in a manner similar to what was presented by Rishbeth and Müller-Wodarg (2006). We compute the January AI using data from that month and from the mean of the preceding and following July. Each July point is computed using data from that month as well as an the mean of the preceding and following

5   January. This was done to smooth out variations in the AI that would otherwise occur due to solar activity changes within six months. Data is presented for the Wallops–Hobart (green), Wakkanai–Port Stanley (blue), and Kodaikanal–Huancayo (red) station pairs to recreate Figures 3 and 4 from Rishbeth and Müller-Wodarg (2006). We were unable to find sufficient data for their fourth example, the Slough–Kerguellen station pair, and thus, it was replaced by an additional station pair, Poiters–Christchurch (black). Results are presented using data from the ionosonde observations (upper left) and the prediction from the IRI model




using $IG_{12}$ (upper right), IG (lower left), and $IG^{NS}$ (lower right) as the solar cycle inputs. For visual convenience, we re-plot the AI using ionosonde data (dashed lines) on each of the IRI-predicted AI plots.

The solar cycle variation of AI from both the observations and IRI compared well with Figures 3 and 4 from Rishbeth and Müller-Wodarg (2006).The observed AI is positive for all four station pairs with the Poiters–Christchurch pair showing the

greatest value. The observed AI tended to increase in value around the years 1970, 1980, and 1990; years corresponding to periods of high solar activity. This indicates that the AI varies with solar activity (Rishbeth and Müller-Wodarg, 2006). These results also show that the IRI-predicted AI, using $IG_{12}$ does not demonstrate a this solar cycle variation. The values are nearly constant for all years presented and do not follow the observations. Confirming that this particular aspect of the IRI model predictions has not been improved since Rishbeth and Müller-Wodarg (2006) first indicated this particular issue. This also

establishes a context by which to interpret improvements to the model predicted AI solar cycle variation when using either of the monthly indices in place of $IG_{12}$.

Using either of the monthly indices, IG or $IG^{NS}$, in IRI improved the predicted AI solar cycle variability however, using $IG^{NS}$ the best agreement of the predicted AI with the observations. When using the monthly IG, the predicted AI increases with solar activity similar to the observations. But for low solar activity years, the predicted AI decreases to considerably lower

values than the observed AI (and is negative at times). Only the observed AI from the Wakkanai-Port Stanley station pair demonstrates this exaggerated solar cycle variability. Using $IG^{NS}$ to predict the AI showed the best agreement with the solar cycle variation of the observed AI. The predicted AI, using $IG^{NS}$ increased with solar activity, as the observed AI for three of the station pairs. The predicted AI uing $IG^{NS}$ did not underestimate the observed AI at low solar activity as with IG. The exception was Wakkanai–Port Stanley, where the predicted values were higher than the observed values during the low solar

activity periods. We suspect that this is related to the unique NmF2 climatology at Port Stanley, which is known to be different from the rest of the Southern Hemisphere (Richards, 2001). We comment on this further in the discussion section.

**Table 7.** Correlation coefficients of the AI from observations with the AI from IRI.

| Station-Pair | $IG_{12}$ | IG | $IG^{NS}$ | NPTS |
|---|---|---|---|---|
| Wa-Ho | 0.10 | 0.59 | 0.84 | 47 |
| Po-Ch | 0.02 | 0.29 | 0.73 | 22 |
| Eg-No | — | — | — | 2 |
| Wa-Po | 0.06 | 0.59 | 0.02 | 15 |
| Ak-To | 0.60 | 0.96 | 0.83 | 3 |
| Ma-Ra | 0.54 | 0.54 | 0.56 | 12 |
| Ko-Hu | 0.90 | 0.80 | 0.84 | 7 |
| Bo-Ho | 0.21 | 0.58 | 0.71 | 63 |
| At-Ca | 0.00 | 0.66 | 0.31 | 12 |
| Gi-Gr | 0.91 | 0.79 | 0.88 | 4 |
| Gi-He | — | — | — | 2 |
| At-Gr | 0.01 | 0.21 | 0.54 | 15 |
| At-He | 0.85 | 1.00 | 0.79 | 4 |
| Average: | 0.38 | 0.64 | 0.70 | |

Table 7 presents the linear correlation coefficient that was computed using the observed AI and the IRI-predicted AI, using the different IG indices. The correlation coefficients range from 0.00 to 0.96. Overall, the IRI-predicted AI values have the greatest correlation with the ionosonde calculated AI when using $IG^{NS}$ as the solar cycle input, on average is 0.70. The

average correlation coefficient when using $IG_{12}$ results in the lowest overall correlation coefficient of 0.38, and using IG gives





a relatively higher correlation coefficient of 0.64. The correlation coefficient is highest for the individual station pairs when using $IG^{NS}$. The Wakkanai–Port Stanley station pair is the exception to this because the correlation coefficient is greatest when using IG (we exclude this from the average) and is very poor when using $IG^{NS}$.

## 3.2 The Solsticial Variation

5    Figure 5 presents the solstice differences, $S_{foF2}$, from the ionosonde observations (black) and IRI using $IG_{12}$ (red), IG (purple), and $IG^{NS}$ (blue). Data is plotted as a function of each station's geographic latitude. We include data from the high solar activity level for each station in the left column. In lieu of missing observations, we use data from the high moderate solar activity level. We present the deep low solar activity level data in the right column. In lieu of missing data we instead present observations from the low solar activity level. The top row shows northern latitude stations while the bottom row shows southern latitude

10   stations. Table 8 presents a summary of the average $S_{foF2}$ values under high and low solar activity conditions separated by Northern and Southern hemispheres (using the different IG indices).





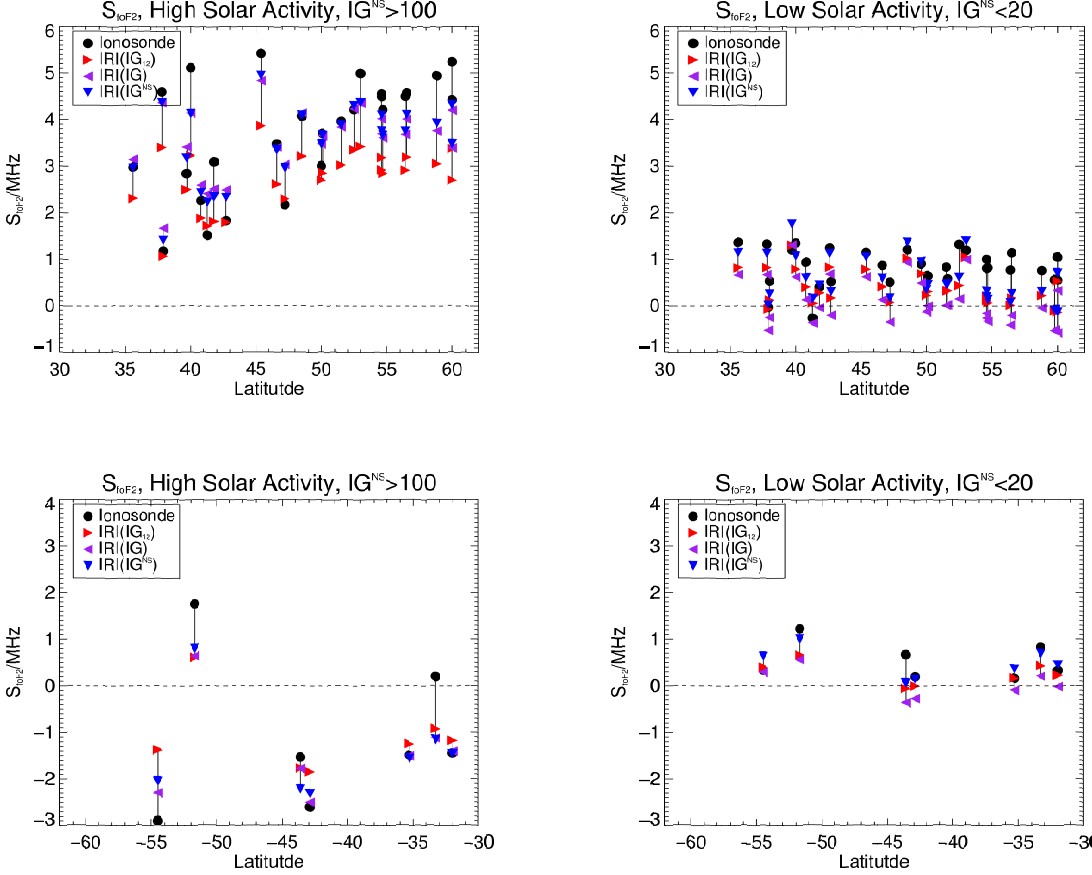

**Figure 5.** Solstice differences, $S_{foF2}$ plotted as a function of each station's geographic latitude, with the Northern hemisphere shown on the top row. The left column presents high solar activity level results; moderate high results are presented in lieu of the missing data, $IG^{NS}_{12} > 100$. The right column presents the deep low solar activity results; low solar activity is shown in lieu of the missing data.

Under high solar activity levels, the $S_{foF2}$ values from the observations vary from -3Mhz to 5.5Mhz. Over the Northern Hemisphere, $S_{foF2}$ is always positive while over the Southern Hemisphere, $S_{foF2}$ is negative. This indicative of the winter anomaly over both hemispheres. The exceptions are two stations, Port Stanley and Grahamstown, which do not show the winter anomaly as the $S_{foF2}$ values are instead positive. The magnitude of $S_{foF2}$ is greater over the Northern Hemisphere than the Southern Hemisphere and is as great as 5.6Mhz (in magnitude) over the Northern Hemisphere but only as great as 3.1Mhz (in magnitude) over the Southern Hemisphere. This indicates that the solsticial variation is more pronounced over the Northern Hemisphere. The IRI model predictions tends to underestimate the solstice differences under high solar activity conditions. The overall the IRI predictions are 26% lower than the observed values when using $IG_{12}$ as solar cycle input. The IRI predictions differ by 1.0Mhz over the Northern Hemisphere and 0.5Mhz over the Southern Hemisphere, on average. This indicates the underestimation of the solsticial variation by IRI is to a greater degree over the Northern Hemisphere than




the Southern hemisphere. Using the monthly indices, IG and $IG^{NS}$, results in average underestimations of 4% and 3% of the observed soslticial variation, respectively. The predicted $S_{foF2}$ on average differs from the observed quantities by 0.20Mhz over the Northern Hemisphere and by 0.10MHz over the Southern hemisphere. This indicates that the monthly indices improve predictions of the hemispheric asymmetry and magnitude of the solsticial variation at high solar activity conditions.

At low solar activity, the magnitude of $S_{foF2}$ varies between 0Mhz and 1.4Mhz. As the value of $S_{foF2}$ is positive for both hemispheres, the winter anomaly is only observed over the Northern hemisphere. $S_{foF2}$ is slightly more pronounced over the Northern hemisphere, presenting an average of 0.89Mhz over the Northern hemisphere and 0.56Mhz over the Southern hemisphere. This is indicative of the annual anomaly at low solar activity. Similar to the high solar activity case, the IRI model underestimates the observed $S_{foF2}$ on average by 69% or by 0.50Mhz over both hemispheres. We note that the percentage

differences are high under low solar activity conditions because the $S_{foF2}$ values used are much smaller in magnitude causing the percent difference causing to be more sensitive to smaller changes in $S_{foF2}$. Using the monthly IG as the solar input worsens the model predictions of $S_{foF2}$, differing on average by 88% (0.80Mhz). Over the Northern hemisphere, the predicted sign of $S_{foF2}$ is often negative, although the observations show a positive $S_{foF2}$. The model predictions are closest to the observations when using $IG^{NS}$, which differ on average by 0.30Mhz over the Northern hemisphere and by 0.05Mhz over

the Southern hemisphere, an overall difference of 22%. This indicates that $IG^{NS}$ is optimal for predictions of $S_{foF2}$ and it's hemispheric asymmetry at low solar activity. Regardless, using $IG^{NS}$ improves predictions of the solsticial variation.

**Table 8.** Average $S_{foF2}$ Separated by Hemisphere

| Average $S_{foF2}$ | High SA | | Low SA | |
|---|---|---|---|---|
| | NH | SH | NH | SH |
| Ionosonde | 3.74 | -1.99 | 0.89 | -0.56 |
| IRI[$IG_{12}$] | 2.74 | -1.48 | 0.39 | -0.1 |
| IRI[IG] | 3.54 | -1.89 | 0.11 | -0.09 |
| IRI[ $IG^{NS}$] | 3.52 | -1.92 | 0.56 | -0.52 |

## 4   Discussion

The results of our analysis indicate that the IRI, which uses the $IG_{12}$ index as the solar cycle input by default, underestimates the magnitude of the annual anomaly by an average of 33%. The solsticial variation analysis indicated that IRI underestimates

the overall magnitude of the foF2 solsticial variation and does not fully predict the it's observed hemispheric asymmetry. This corresponds with the underrepresented predictions of the magnitude of annual anomaly by IRI. Using the monthly IG index as solar cycle input caused unrealistic predictions of the magnitude of the annual anomaly under low solar activity conditions. The model predictions of the magnitude of the annual anomaly are improved when the $IG^{NS}$ index is used for solar cycle input; predictions now underestimate the observed anomaly by 6%. This corresponded with improved predictions of both the

magnitude and hemispheric asymmetry of the solsticial variation when using the $IG^{NS}$ as solar cycle input to the IRI. Thus, our results indicate that a monthly hemispheric index is necessary to improve model predictions of the annual anomaly.





The exception to what our results suggest is the Port Stanley–Wakkanai station pairing where the IRI predictions overestimated the annual anomaly when using the $IG^{NS}$ index as solar cycle input. Torr and Torr (1973) noted that the solsticial variation observed at Port Stanley differed from the remainder of the Southern Hemisphere by showing a greater magnitude, with very little variation with solar activity, and higer values in January than July. Richards (2001) also noted this unique

climatology. Both authors suggested that this phenomenon may be related to the vicinity of Port Stanley to the South Atlantic Anomaly, where gradients in the geomagnetic field are sufficiently larger than those on other parts of the globe to alter the transport properties of the neutral atmosphere. We suspect that it is difficult to use the $IG^{NS}$ to accurately describe this AI because this index describes the average behavior of the Southern Hemisphere, from which the ionospheric climatology at Port Stanley departs.

The underestimation of the magnitude of the annual anomaly (and solsticial variation) using the IRI model is related to underlying the formulation of the IG and $IG_{12}$ indices. Using $IG_{12}$ for solar cycle input data resulted in an overall 31% underestimation of the solsticial variation using the IRI model under high solar activity conditions. The use of either monthly indices (IG and $IG^{NS}$) improved upon this, underestimating the observed solsticial variation under high solar activity conditions by 5%, on average. This suggest that the 12-month averaging of the IG index diminishes the magnitude of the predicted solsticial

variation. The same holds true under low solar activity conditions with one notable exception.

Predictions of the solsticial variation worsened when using the standard IG index under low solar activity conditions. In their study, Brown et al. (2017) showed that using the standard monthly IG caused unrealistic predictions when the CCIR-66 foF2 was used to specify the foF2 over the Northern hemisphere during low solar activity periods. This indicates that the global averaging of the IG index is not optimal for specifications over both the Northern and Southern hemispheres at low solar

activity. They also showed that, because the IG index is computed using the CCIR-66 foF2 model, monthly values of the index are incompatible with the URSI-88 foF2 model. This explains why predictions of the solsticial variation presented in this study worsened over the Northern Hemisphere under low solar activity conditions when using the standard monthly IG index. Brown et al. (2017) also showed that although the $IG_{12}$ index is also computed using the CCIR-66 foF2 model, the 12-month averaging decreases the magnitude of the unrealistic index values which would cause the unrealistic foF2 predictions. Therefore, the use

of a 12-month averaged index in both foF2 models is acceptable and averaging is necessary to mitigate the problems associated with model–index incompatibility.

Improvements to the IRI predictions of the annual anomaly are related to three qualities regarding our index, $IG^{NS}$. First, the $IG^{NS}$ is not averaged over 12 months, which improved predictions of the magnitude of the solsticial variation when using the IRI model. The index is computed using the URSI-88 foF2 model, which resolves problems associated with the model–index

incompatibility introduced by the inclusion of a monthly index in the IRI in place of a 12-month averaged index. Finally, $IG^{NS}$ is averaged for each hemisphere and not over the whole globe, which improves predictions of ionospheric variations particular to a particular hemisphere. For these reasons, the model predictions of the annual anomaly are improved.

Because improvements to the IRI model predictions of the annual anomaly were obtained with the inclusion of the hemispheric $IG^{NS}$ ionospheric index, we can infer attributes the anomaly's underlying mechanisms. During low solar activity

periods, solar EUV, F10.7, and geomagnetic activity, parameters that typically drive ionospheric variations, are at their low-



est levels. Therefore, estimations of the annual anomaly under low solar activity conditions will primarily be affected by Sun–Earth distance variations and other ionospheric background processes that are typically obscured by geophysical conditions. These ionospheric processes, which include geomagnetic, meteorological, and dynamical processes, are inherently described by ionospheric indices, unlike solar indices (such as the F10.7 and sunspot number), which can only account for

solar irradiance variations. Therefore, because predictions of the annual anomaly using the IRI model were improved by using a hemispheric ionospheric index, under low solar activity conditions, we infer that the underlying processes that drive the anomaly also vary over each hemisphere.

Rishbeth and Müller-Wodarg (2006) made one of the initial suggestions that the annual anomaly may be the result of some differences between the Northern and Southern hemispheres rather than a difference in conditions between the solstices. The

reasons for which the ionosphere is not the same over both hemispheres are well documented. These reasons include differing configurations of the geomagnetic fields, varied auroral electrojet indices, and a persistent difference in auroral hemispheric power (Mikhailov and Perrone, 2015, and references therein). Regarding the geomagnetic configuration, model simulations by Zeng et al. (2008) showed that the offset of the geomagnetic center from the geographic center can account for 40% of the annual anomaly. However, understanding the dynamical changes to both the neutral and ionized atmosphere, as driven

by changes in the geomagnetic field configuration, is unclear and requires further investigation. Liu et al. (2007) suggested that the field-aligned transport of the ionospheric plasma along the geomagnetic field by the neutral wind differs over the two hemispheres. They showed that the wind tends to uplift the ionosphere over the Southern Hemisphere more so than the Northern Hemisphere in January. This enhances the ionospheric electron density in January over the Southern hemisphere (relative to July), contributing to the the annual anomaly. This mechanism has not been tested self-consistently. Another possibility is

role of the vertically-propagating gravity waves from the lower atmosphere to the F2 region of the ionosphere and the annual anomaly. The momentum and heat deposition by gravity waves in the F-region of the ionosphere has been shown by Yiğit et al. (2008) and Yiğit et al. (2009) and Miyoshi et al. (2014) to play a significant role in the thermosphere's mean climatology. Global climate simulations by Yiğit and Medvedev (2017) suggest that the effects of gravity waves on the neutral atmosphere differ over the northern and southern hemispheres. There is no literature that directly explores the effects of gravity waves on

the annual anomaly. Although the extent to which these physical processes contribute to the annual anomaly is not clear, the success of the IG$^{NS}$ index at improving IRI predictions of the magnitude of the anomaly suggest that further study of this hemispherically-asymmetric processes should be the topic of future investigations.

## 5  Summary and Conclusions

In this work, we presented an investigation of the IRI predictions of the ionospheric annual anomaly and solsticial variation

using new ionospheric indices as solar cycle input. We compared observational data against IRI model predictions using either the standard monthly IG, it's 12-month average ,IG$_{12}$ or IG$^{NS}$–the hemispheric IG index introduced by Brown et al. (2017)–as solar cycle input. Through retrospective analysis, it was shown that the IRI, which currently uses the IG$_{12}$ index for solar cycle data input, underestimates the magnitude of the observed annual anomaly by 33%. This is related to an underestimation of both





the magnitude and hemispheric asymmetry of model predictions of the solsticial variation. The underestimated magnitude is caused by using a 12-month averaged IG index instead of a monthly index.

Using the monthly IG index resulted in a 37% underestimation of the annual anomaly. Although the monthly IG index improved predictions of the annual anomaly under high solar activity conditions it worsened predictions under low solar activity
conditions. This is attributed to both the global averaging process used to compute IG as well as the model-index incompatibility of the index with the IRI model. Using $IG^{NS}$ improves the model predictions under both high and low solar activity conditions, underestimating observed values by 6%, on average. The use of $IG^{NS}$ improves both the predicted magnitude and predicted hemispheric asymmetry of the solsticial variation under high and low solar activity conditions because the $IG^{NS}$ index is computed for each hemisphere. Therefore, because predictions of the annual anomaly by IRI were improved by using
a hemispheric ionospheric index, under low solar activity conditions, we infer that the underlying processes that drive the anomaly also vary over each hemisphere.

*Author contributions.* SB mainly did data collection, analysis and initial write-up. All authors participated in the writing and all commented on the paper.

*Competing interests.* No competing interests are present.

*Acknowledgements.* The authors would like to thank the National Geophysical Data Center (NGDC) and the Global Ionospheric Radio Observatory (GRO) for providing the ionosonde station data used in this paper. The authors would also like to extend a thank you to all referees for help in evaluating this paper.





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
