# Peer review of "Improvements to Predictions of the Ionospheric Annual Anomaly by the International Reference Ionosphere Model"

_Annales Geophysicae, 2018_

## Referee Comment (RC1) · Anonymous Referee #1 · 25 Sep 2018

This study uses a newly conceived method of calculating hemispheric adjustments to the IRI URSI foF2 maps to investigate the model's capacity to represent the Annual Anomaly. The authors illustrate that the IRI performs better at reproducing this anomaly when using hemispheric indices versus using a single ionospheric adjustment index (IG). I have a few concerns regarding the concept of this study. The method used here to improve the IRI is essentially a two-pixel assimilation scheme (one for each hemisphere). It is hardly a question that such a scheme would improve the model's capacity to represent an anomaly that depends on the differential behaviour of the two hemispheres, especially when the stations you assimilated (i.e. used to develop IGNS) are the same ones being used to evaluate this phenomenon. The authors argue

that the improved performance through the use of separate hemispheric indices over a sole global index has physical connotations and can be related to differential solar activity scaling between the hemispheres. This conclusion, however, relies on the assumption that the errors in the URSI foF2 maps are related to physical behaviour and not the result of simply issues in the original fitting of the URSI foF2 maps, which were plagued by hemispherically asymmetric datasets, resulting in more artificial data being used in the Southern Hemisphere. The issue of data coverage and breadth has always been a barrier to the use of the IRI for teasing out physical conclusions and I believe that remains the case with this study. Regardless, in any assimilation-like scheme, increased flexibility in the basis set (i.e. allowing for two indices rather than one) would, by construction, improve performance with respect to the fitting dataset and thus improve the model's ability to represent the Annual Anomaly. Until the cause of the IRI's original AI underestimation is properly identified, drawing physical interpretations is largely conjecture.

In addition to the above I also have concerns regarding the ultimate utility of these hemispherical adjustments to the model:

1) Is this considered a potential replacement to the current methodology in the IRI? Using a hemispheric adjustment would likely create significant artificial gradients at the boundary between the hemispheres, making this approach unusable for many applications that rely on the IRI, such as trans-equatorial HF propagation modeling. 2) What is the advantage of this method over already available products, such as IRTAM, the real-time IRI? The IRTAM is presumably a far more robust assimilation scheme and is readily available at the moment. Where does your method fit? 3) One of the features of the current IG12 index is its stability to be forecasted. How would you implement this methodology in the IRI to retain the forecast capability?

I realize that some of these questions are more pertinent to the author's previous publication, but they are nonetheless relevant to this application of the method.

[Figure]

Based on the above concern regarding the legitimacy of the authors' physical interpretation of empirical model behaviour, I recommend that this study be rejected. There are valuable components of this study that may be worth publication; however, the main thesis of this document is far from convincing without significant additional investigation; in fact, the main point of the document may be better illustrated without the use of the IRI and instead solely focusing on observations.

Other Major Comments:

1) Figure 1 – Do you have copyright approval to reproduce this figure? If so, please state so and list the publisher. 2) Figure 2 – There are significant holes in the station distribution. Perhaps comment on how this may affect your results. 3) Table 4 – Do you have statistical error information for the values in this table? Are these statistically significant? 4) Figure 4 – Can you comment on the differences between your top left figure here and that from Rishbeth and Muller-Wodarg (2006)? There appear to be differences in the ionosonde-derived AI values for the same pairs. Were you unable to acquire the same ionosonde data or are there processing differences between your study and theirs? This would also somewhat highlight the need to have some sort of error measure associated with AI values. 5) Regarding the incompatibility between the IG from CCIR and using it for the URSI maps – Why would you not just recalculate a monthly IG index for the URSI maps, as you have done for the IGNS index? This way you would be able to definitively define where the errors are coming from.

Minor Comments:

Page 2, Line 15 – Please cite your previous paper regarding the method of determining the IGNS here.

Page 5, Line 1 – "utilized" -> "using" or "utilizing"

Page 6, Last Line – AP -> Are you referring to Ap, the daily arithmetic mean of the three hourly ap index, or are you referring to the three hour ap index itself. A bit semantic

being a capitalization difference, but these two indices are often confused.

Table 8 – Low SA -> move to right.

Page 18, Line 10-12 -> Perhaps mention the differential behaviour of the north and south PC indices here as well?

Page 19, Line 16 – GRO -> GIRO

---

## Referee Comment (RC2) · Anonymous Referee #2 · 25 Sep 2018

The manuscript attempts to improve the annual anomaly representation in the most acknowledged empirical model of ionosphere, IRI. Although the goal is very important, the study is flawed due to various mathematical and notational errors. The paper is also full of grammatical and syntax errors. Here is a list of major problems with the paper: 1) The 'prediction' is a mathematical operation where future values of a discrete-time signal are estimated as a function of previous samples. Here, the study is retrospective and deterministic in the sense that a preset algorithm with defined coefficient set is run with different index values. Thus, this operation cannot be considered as 'prediction'. Therefore, the title and the wording in the text should be modified. 2) The definition of AI and the explanation given under the equation 1 are highly problematic. The mean

[Figure]

M is defined to be the sum of two numbers which is wrong. The equation should be corrected to reflect the proper implementation. 3) The implementation of 'AI' is not clear. The explanation mentions that 'The AI is computed by using an average of the NmF2 from both Northern and Southern hemisphere which have similar geomagnetic latitudes . . .' Yet, equation 1 does not indicate any averaging over the stations in both hemispheres. Further in the text, AI is applied to station pairs during daylight hours and for the months of January and July for a set of years. This operation is not reflected in equation 1. The definition of AI should be given properly to reflect the full intent and computation. In its present state, it cannot be accepted. 4) The explanation for the interpretation of the value of AI and the example given are also wrong. 5) The information on the background literature is not given properly. Figure 1 which is taken from another paper, may have copyright issues. It is not clear how the AI index is applied to station pairs and what those legends on the subplots mean. The authors clearly mention that the application in the Rishbeth and Muller-Wodarg (2006) paper did not provide any satisfactory explanation to their results and it has 'reliability' issues, yet they adopted their line of computation of AI index between the station pairs! This is a contradiction in itself. 6) In the official site of IRI, irimodel.org, IG index is mentioned to be an ionospheric index not a 'solar cycle input'. The authors should clearly define what they mean by solar cycle input. 7) Apparently, the IRI model utilizes a set of coefficients and index values in the computation of NmF2 and foF2 for a user defined date, hour and location. The model uses IG12 from IGRZ.dat file. Since the model aims to produce hourly monthly medians, 12 month running median of IG is automatically input from the data files. If the user wishes to update the value, he or she can input it separately at the time of run. How did the authors prepare the index set for IG and IGNS? Since they are not available in a format that can be input automatically in the online version of IRI-2016, did the authors run the model offline in the Fortran version? 8) According to the information given in the introduction section, IGNS is developed using 50 ionosonde station so it is an ionospheric index more than a solar cycle input. In Figure 2, there is a map of the world with black dots indicating the ionosonde stations

used in the study (and ionosonde is misspelt!). Then, there is Table 2 which lists a set of stations that are used in the study. Most of the stations indicated on the map are not listed in the Table and some stations such as Eglin, Florida and Huancayo, Peru are not on the map! The pairing of the stations are also flawed. The stations are paired according to not only north-south hemispheres but also east-west hemispheres! The geomagnetic coordinates are not taken into account and station in Virginia, USA is paired with another station in Tasmania. If the pairing is necessary, at least magnetic conjugates and local daylight hours should be considered. For example, the stations in Japan can be matched with those in Australia and New Zealand. The stations in Europe can be matched with those in South Africa. If the authors have some other mechanism in mind, they have to explain this in a better way. Otherwise, this kind of station pairing does not make sense at all. Taking the 'mean' of latitude of two stations in two different hemispheres do not make any sense either mathematically or physically. 9) The application of AI to a station pair, for daylight hours, for the months of January and July and for a set of years and 'averaging' should be clearly given in a mathematical equation with proper notation. 10) The authors should know by now that the units are never written in a mathematical equation. The unit of NmF2 is not 1/metercube but el/metercube. The unit of frequency is indicated by Hz not hz. The units are never written in italics. There should always be one blank space between the number and the unit. The asterisk is not a proper mathematical notation for multiplication. 11) Equation 3 does not represent the parameters of the application. For one ionosonde station that computes foF2 every 15 minutes, how can there be one value for whole month of January or July? 12) Figure 3 is also very unclear. The horizontal line for 80 is missing. The main problem is that the plot is prepared for the 12 month running mean of IGNS not IGNS itself. Which input data file did the authors use in this study? The years chosen for various levels of solar activity are given in Table 3. According to these information, the study covers the years from 1970 to 2014, yet in the rest of the paper, the results are provided for 1970 to 1990 (such as Figure 4) or the years are not mentioned at all. For a list of 8 years for low activity years,

there is only one value in the tables. What happened to the data? The criterion for less than 10 percent is not clear. 13) In Tables 4, 5 and 6, there are two stations and one 'Iono' value. How is this possible? How did the authors compute the values? The caption of the titles mention that the values are AI, yet the column titles indicate that they are IRI(IG) or Iono. The 'average' and 'average*' operations are not clear. Why did the authors include station pairs with no data into the tables? 14) In Table 6, the conclusions drawn are wrong. For 7 station pairs that have data, IG input matched 4 of these, whereas IGNS matched only 2. So IG is a better input for high solar activity years. 15) Figure 4 does not include years from 1990 to 2014. It is not clear why all the stations are not given? What is the meaning of diamonds? Some lines have them and some does not. If they are the years of computation, this information does not match Table 3. The subplots are too crowded. 16) The information on the computation of AI on page 12 is not clear at all. All mathematical computations should be clearly indicated with proper notation and equation numbers. 17) I have reservations for the 'missing data replacement' as done in the manuscript. 18) Table 7 does not make sense at all. The correlation coefficients are computed for which data sets? What is NPTS mean? Is the correlation biased or unbiased? For what years and for which solar activity level? 19) Figure 5 is also another mystery. Even the labels are misspelt. What does the vertical bars or lines represent? And so on... 20) Table 8 is wrong. 21) Since the equations for computation of AI, averaging of AI and SfoF2 are totally unclear and may possibly contain significant physical and mathematical errors, none of the comments in the discussion or the drawn conclusions are reliable. 22) The paper is full of notational, grammatical and mathematical errors. The authors should start using at least a spell-checker and technical reviewer to edit the paper. There are syntax errors and singular-plural errors. There are many incidences of two verbs are used in the same sentence. 23) The authors should stop 'suspecting'. 'Suspect' is not part of scientific terminology. 24) There is no references for IGRF model and some of the data sources are not acknowledged.

The paper cannot be accepted in its present form.

---

## Author Comment (AC1) · 21 Nov 2018

The authors would first like to extend their gratitude to referee 1. We appreciate the referee's insight, time and diligence in reviewing this manuscript. What follows is a response to the referee's comments. We tried to break the referee's comments into sections and respond directly. Thank you

———————————

———————————

[referee] This study uses a newly conceived method of calculating hemispheric adjustments to the IRI URSI foF2 maps to investigate the model's capacity to represent the

[Figure]

Annual Anomaly. ...... increased flexibility in the basis set (i.e. allowing for two indices rather than one) would, by construction......Until the cause of the IRI's original AI underestimation is properly identified, drawing physical interpretations is largely conjecture.

[response] We agree and have changed the text as follows:
We have removed all text concerning physical interpretations from the end of the discussion, including aspects of it in the introduction. This text includes pages 17 lines 33-35 and page 18 lines 1-27.
We have replaced this text with discussion:
"Improvements to the IRI predictions of the annual anomaly are related to three qualities regarding our index, IGNS, which improve upon the other indices.
—the index is computed separately for Northern and Southern hemispheres
—the index is not averaged over 12-months
These first two qualities are the most important. These qualities are consistent with the observed features of the annual anomaly: the anomaly occurs over time scales less than 1 year and causes the ionosphere's climatology over the Northern hemisphere to differ from the Southern hemisphere. Thus observations of the annual anomaly drive the improvements needed to improve IRI's description of the anomaly.
— the index is computed using the URSI-88 foF2 model.
"This addresses the problems associated with model-index incompatibility raised by Brown et al. (2017). Undoubtedly, the reliability of IRI's description of the annual anomaly is related to the original fitting of the CCIR and URSI foF2 maps, which were computed using an uneven hemispherical distribution of observational data with a strong bias towards the Northern hemisphere. Using separate hemispheric indices to drive the CCIR and URSI models helps to overcome some of this hemispheric bias and results in a better description of the annual anomaly."

——————————————
——————————————

[referee] Is this considered a potential replacement to the current methodology in the

IRI? Using a hemispheric adjustment would likely create significant artificial gradients at the boundary between the hemispheres, making this approach unusable for many applications that rely on the IRI, such as trans-equatorial HF propagation modeling.

[response] The reviewer makes a very good point that needs to be addressed by future studies. Most likely a smoothing algorithm in a transition region could help to overcome this problem. This is outside of the scope of this study that primarily wanted to assess the benefit of hemispheric indices in improving IRI's description of the annual anomaly.
* * ** * *
[referee] What is the advantage of this method over already available products, such as IRTAM, the real-time IRI? The IRTAM is presumably a far more robust assimilation scheme and is readily available at the moment. Where does your method fit?

[response] Both the standard IRI and IRTAM version of IRI are important ionosphere models, which are needed for a number of applications. Efforts to improve both should continue. The IRI is recognized by the International Standardization Organization (ISO) as the ISO standard for the ionosphere and its full functionality is freely available to the international community. The full functionality of the IRTAM is available at a premium, a limited version is available freely. Our index improves the IRI by reducing foF2 model prediction errors. This is also of benefit to IRTAM, which uses IRI for its initialization. With a more realistic first guess IRTAM will more quickly converge to represent real-time conditions. IRI is also critical to IRTAM as the background ionosphere for regions where no data are available.

To emphasize this point in the text we have added the following to the first paragraph of the INTRODUCTION (page 2 line 4):

"….is widely used for the empirical specification of the ionosphere. Recent efforts have concentrated on assimilating data into IRI to more accurately represent real-time

conditions. The IRI Real-Time Assimilative Mapping (IRTAM) is a first step towards this goal assimilating ionosonde data for foF2 and hmF2 into IRI. Both the standard IRI and IRTAM version of IRI are important ionospheric models, which are needed for a number of applications. The IRI is recognized by the International Standardization Organization (ISO) as the ISO standard for the ionosphere and its full functionality is freely available to the international community. Efforts to improve the IRI is of benefit to IRTAM, which uses IRI for its initialization. With a more realistic first guess IRTAM will more quickly converge to represent real-time conditions. IRI is also critical to IRTAM as the background ionosphere for regions where no data are available. Efforts to improve both models should continue. There are other efforts underway as well as described by Bilitza et al. (2011, 2017)."

And will update the bibliography to include the following citations:

Bilitza D., L.-A. McKinnell, B. Reinisch, and T. Fuller-Rowell, The International Reference Ionosphere (IRI) today and in the future, J. Geodesy, 85:909–920, DOI 10.1007/s00190-010-0427-x , 2011

Bilitza, D., D. Altadill, V. Truhlik, V. Shubin, I. Galkin, B. Reinisch, and X. Huang (2017), International Reference Ionosphere 2016: From ionospheric climate to real-time weather predictions, Space Weather, 15, 418–429, doi:10.1002/2016SW001593.

——————————

——————————

[referee] One of the features of the current IG12 index is its stability to be forecasted. How would you implement this methodology in the IRI to retain the forecast capability?

[response] The reviewer makes a very good point. A monthly index provides better results when used for retrospective modelling, e.g., creating a re-analysis data set combining existing data with model inputs. To retain the forecast capability, we commend to use a, a 12-month averaged IGNS index.

————————————

————————————

[referee] 1) Figure 1 – Do you have copyright approval to reproduce this figure? If so, please state so and list the publisher.

[response]We do not have the copyright approval. This figure was removed from the body of the manuscript.

————————————

————————————

[referee] 2) Figure 2 – There are significant holes in the station distribution. Perhaps comment on how this may affect your results.

[response] The reviewer makes a good point. We will include the following text in the methodology:
(page 7 line 6) "The station distribution will not have an effect on our results regarding the annual anomaly because we always use station paired with one from the Northern hemisphere and one from a magnetically conjugate location in the Southern Hemisphere."

————————————

————————————

(page 7 line 13) "Because there are more stations in the Northern hemisphere than in the Southern hemisphere our results for the Solstitial variation are more statistically significant for the Northern hemisphere."

————————————

————————————

[referee] 3) Table 4 – Do you have statistical error information for the values in this table? Are these statistically significant?

[response] We do not have a meaningful statistic for the station pair AI. All literature concerning the AI index calculation also does not present a statistic for the AI. Perhaps due to the small number of data points required for a single AI calculation.

We have begun to investigate this further. At present, we are considering to include an RMS

——————————

——————————

[referee] 4) Figure 4 – Can you comment on the differences between your top left figure here and that from Rishbeth and Muller-Wodarg (2006)? There appear to be differences in the ionosonde-derived AI values for the same pairs. Were you unable to acquire the same ionosonde data or are there processing differences between your study and theirs? This would also somewhat highlight the need to have some sort of error measure associated with AI values.

[response] We agree, we will add the following text to the first paragraph of section 3.1.1:
(page 12 line 1): "Data are presented for the Wallops–Hobart (green), Wakkanai–Port Stanley (blue), and Kodaikanal–Huancayo (red) station pairs for the years 1970-1990, a subset of our full data record. This subset of our data record is presented in order to recreate Figures 3 and 4 from Risbeth and Muller-Wodarg (2006) and perform a direct comparison with their work. We emphasize three contributing factors which will cause deviations of the charts presented in this work and those presented by Rishbeth and Muller-Woodarg. First, we were unable to find sufficient data for their fourth example, the Slough–Kerguellen station pair, and thus, it was replaced by an additional station pair, Poiters–Christchurch (black). Second, the charts presented in this work use monthly median observations in the range of 10LT to 14LT. Rishbeth and Woodarg (2006) indicated using observational data from 12LT. And thirdly, our data were obtained from both SPIDR and GIRO data repositories, Rishbeth and Muller-Wodarg did

not use data from the GIRO repositories. We do not expect these contributing factors to significantly affect our results. "
* * ** * *
[referee] 5) Regarding the incompatibility between the IG from CCIR and using it for the URSI maps – Why would you not just recalculate a monthly IG index for the URSI maps, as you have done for the IGNS index? This way you would be able to definitively define where the errors are coming from.

[response] This manuscript emphasizes a comparison of IRI predictions using our new index, IGNS, with predictions using the official (CCIR-based) IG and IG12 indices. Currently IRI uses the CCIR-based IG indices with the CCIR maps as well as the URSI maps. So the use of an URSI-based IG index as proposed and studied by Brown et al. (2017) will result in additional improvements of IRI predictions. We point out that this model-index incompatibility was problematic for low solar cycle conditions. Our index demonstrated improvements for all solar cycle levels.
* * ** * *
[referee] Minor Comments:

Page 2, Line 15 – Please cite your previous paper regarding the method of determining the IGNS here.
Page 5, Line 1 – "utilized" -> "using" or "utilizing"
Page 6, Last Line – AP -> Are you referring to Ap, the daily arithmetic mean of the three hourly ap index, or are you referring to the three hour ap index itself. A bit semantic being a capitalization difference, but these two indices are often confused.
Table 8 – Low SA -> move to right.
Page 18, Line 10-12 -> Perhaps mention the differential behaviour of the north and

south PC indices here as well?
Page 19, Line 16 – GRO -> GIRO

[response] We have addressed the listed minor points.
We used the daily arithmetic mean of the Ap. This is now clarified in the text. We have included the following additional text (page 2, line 15) "These adjustments improve the predictions of temporal and spatial variations in foF2 that are made by the IRI model. Please refer to Brown et al. (2017) for a full description of the calculation of IGNS."
* * *
Table 4.8: Average $S_{foF2}$ Separated by Hemisphere

| Average $S_{foF2}$ | High SA | | Low SA | |
|---|---|---|---|---|
| | NH | SH | NH | SH |
| Ionosonde | 3.74 | -1.99 | 0.89 | -0.56 |
| IRI[IG$_{12}$] | 2.74 | -1.48 | 0.39 | -0.1 |
| IRI[IG] | 3.54 | -1.89 | 0.11 | -0.09 |
| IRI[ IG$^{NS}$ ] | 3.52 | -1.92 | 0.56 | -0.52 |

**Fig. 1.** Revised Table 4.8

---

## Author Comment (AC2) · 21 Nov 2018

The authors would first like to extend their gratitude to the reviewer. We appreciate the reviewer's insight, time and diligence in reviewing this manuscript. What follows is a response to the referee's comments. We agreed with a majority of the issues/concerns raised by the reviewer and have replied with additional comments, edits and clarification. Thank you.
——————-
——————-

[referee] 1) The 'prediction' is a mathematical operation where future values of a

discrete-time signal are estimated as a function of previous samples. Here, the study is retrospective and deterministic in the sense that a preset algorithm with defined coefficient set is run with different index values. Thus, this operation cannot be considered as 'prediction'. Therefore, the title and the wording in the text should be modified.

[response] We agree with the reviewer's logic, but want to point out that the use of "prediction", as is utilized in this manuscript, is an established practice throughout the ionospheric modeling community. A web search yields more than 40 journal articles that present retrospective comparisons between observations and IRI and use the term "prediction" in the title and throughout the manuscript.

——————-

——————-

[referee] 2) The definition of AI and the explanation given under the equation 1 are highly problematic. The mean M is defined to be the sum of two numbers which is wrong. The equation should be corrected to reflect the proper implementation

[response] The equation presented in consistent with the AI expression presented in Rishbeth and Muller-Woodarg (2006). However the reviewer's point signals a need to clarify a few points in the text. To add clarification in its presentation, we make the following adjustments to the text:

In the introduction
(page 3, line 17), We replace
"The asymmetry index (AI), as introduce by Rishbeth and Muller-Wodarg (2006), commonly used to describe the magnitude of the annual anomaly is defined as: "
(page 3, line 19) Equation 1 is modified: removed "A/M"
(page 3, line 2) remove (A) and remove (M)
(page 3 line 20)

We include a explicit expressions for $NmF2_{NSJan}$ and $NmF2_{NSJuly}$

(page 3, line 21) We also include the following text:
"NmF2$_N S$ $is the average of a pair NmF2 values from magnetically conjugate locations. The subscript 'Jan and Jul indicate if th$

——————-

——————-

[referee] 3) The implementation of 'AI' is not clear. The explanation mentions that 'The AI is computed by using an average of the NmF2 from both Northern and Southern hemisphere which have similar geomagnetic latitudes ...' Yet, equation 1 does not indicate any averaging over the stations in both hemispheres. Further in the text, AI is applied to station pairs during daylight hours and for the months of January and July for a set of years. This operation is not reflected in equation 1. The definition of AI should be given properly to reflect the full intent and computation. In its present state, it cannot be accepted.

[response] As noted in our reply to the Reviewer's point (2) we have included an explicit expression for NmF2$_{NS}$.
We include the following text in the introduction (page 3, line 25):
"The AI has been used to describe the annual anomaly for numerous geophysical conditions and local times. The observational input to AI varies from NmF2 derived from using a pair of ionosonde observations at approximate geomagnetic conjugate latitudes or using NmF2 observed from satellites averaged over lines of constant geomagnetic latitude. Equation (1) is adjusted to fit each case. We reserve the discussion of the implementation of AI for this work or the methodology section"

——————-

——————-

[referee] 4) The explanation for the interpretation of the value of AI and the example given are also wrong.

[response] As noted in our reply to the Reviewer's point (2) we have now rewritten this

part and changed the explanation. The interpretation and explanation on page 3 lines 22-25 are consistent with literature.

——————-

——————-

[referee] 5) The information on the background literature is not given properly. Figure 1 which is taken from another paper, may have copyright issues. It is not clear how the AI index is applied to station pairs and what those legends on the subplots mean. The authors clearly mention that the application in the Rishbeth and Muller-Wodarg (2006) paper did not provide any satisfactory explanation to their results and it has 'reliability' issues, yet they adopted their line of computation of AI index between the station pairs! This is a contradiction in itself.

[response]
We appreciate the author's note on copywrite and will remove the figure, and instead will focus on the conclusions drawn by Rishbeth and Muller-Wood (2006).

We believe the author is referencing page 5, line 4 and onward: The 'Reliability' issues pertain to IRI's specification of the annual anomaly and not the quality or validity of the analysis carried out by Rishbeth and Muller-Woodarg (2006).

——————-

——————-

[referee] 6) In the official site of IRI, irimodel.org, IG index is mentioned to be an ionospheric index not a 'solar cycle input'. The authors should clearly define what they mean by solar cycle input.

[response] We will emphasize this in the introduction by including the following text (page, 2 line 10),
"Currently, these models use the 12-month running mean of the official IG index, IG12 (this ionospheric index is also known as the "global sunspot number") (Liu et al., 1983),

in place of the solar sun spot number, as solar cycle input."

—————-

—————-

[referee] 7) Apparently, the IRI model utilizes a set of coefficients and index values in the computation of NmF2 and foF2 for a user defined date, hour and location. The model uses IG12 from IGRZ.dat file. Since the model aims to produce hourly monthly medians, 12 month running median of IG is automatically input from the data files. If the user wishes to update input it separately at the time of run. How did the authors prepare the index set for IG and IGNS? Since they are not available in a format that can be input automatically in the online version of IRI-2016, did the authors run the model offline in the Fortran version?

[response] We used indices input files with a format identical to IG-RZ.dat with the offline IRI, but using the IGNS values in place of the IG12 values.

—————-

—————-

[referee] 8) According to the information given in the introduction section, IGNS is developed using 50 ionosonde station so it is an ionospheric index more than a solar cycle input. In Figure 2, there is a map of the world with black dots indicating the ionosonde stations used in the study (and ionosonde is misspelt!). Then, there is Table 2 which lists a set of stations that are used in the study. Most of the stations indicated on the map are not listed in the Table and some stations such as Eglin, Florida and Huancayo, Peru are not on the map! The pairing of the stations are also flawed. The stations are paired according to not only north-south hemispheres but also east-west hemispheres! The geomagnetic coordinates are not taken into account and station in Virginia, USA is paired with another station in Tasmania. If the pairing is necessary, at least magnetic conjugates and local daylight hours should be considered. For example, the stations in Japan can be matched with those in Australia and New Zealand.

The stations in Europe can be matched with those in South Africa. If the authors have some other mechanism in mind, they have to explain this in a better way. Otherwise, this kind of station pairing does not make sense at all. Taking the 'mean' of latitude of two stations in two different hemispheres do not make any sense either mathematically or physically.

[response]

The number of stations used for the IGNS index is not stated in this manuscript.

This information has been added to the 2nd paragraph of the INTRODUCTION.

" The IGNS index is developed with 13 stations, IG12 currently uses 4 stations. Please refer to Brown et al., (2017) for a full description of the index computation."

Misspelling was corrected

The stations marked on Figure 2 are the stations used for the solstice analysis. The stations listed in Table 2 are the stations used for the asymmetry analysis. Stations used for the AI analysis need to have data that overlap time periods and are at similar geomagnetic latitudes over both hemispheres. These criteria limit the number of station pairs that can be used I this study thus ,there are less stations listed in table 2 than indicated in Figure 2.

The stations are paired by approximate geomagnetic latitude as determined by the IGRF model. Our stations pairings overlap with several of the pairings used by Rishbeth and Muller-Wodarg (2006) as well as Mikhailov and Perrone (2013). No stations of the same hemisphere were paired. We indicate this in table 2 by also including the geographic coordinates of the corresponding station pairs.

Boulder, Eglin and Millstone Hill were our only North American stations. Perhaps, 'Norfolk" was mistaken for Norfolk, Virginia, USA"? In our table, "Norfolk" refers to Norfolk Island is east of Australia and currently reports to GIRO. We included geographic coordinates in Table 2 for further confirmation. For added clarification we will change

"Norfolk" to "Norfolk Isl" in table 2

We included the following text to line 5 of page 7 of the manuscript to clarify the inclusion of the averaged geomagnetic coordinates: (page 7, line 5) "The station pairings are listed in Table 2. This table list each station pairing by name as well as their abbreviated label, geographic latitude, geographic longitude and geomagnetic latitude. The stations are listed in order of their mean absolute geomagnetic latitudes. The geomagnetic latitudes are specified by the International Geomagnetic Reference Field (IGRF) at a height of 300 km."

——————-
——————-

[referee] 9) The application of AI to a station pair, for daylight hours, for the months of January and July and for a set of years and 'averaging' should be clearly given in a mathematical equation with proper notation.

[response] The equation for the AI index is presented in the introduction is described as using monthly medians input from January and July. In our methodology, we described that for this work, we will use the median of observational data which fell between 10LT and 14LT describe the January and July monthly median values.

——————-
——————-

[referee] 10) The authors should know by now that the units are never written in a mathematical equation. The unit of NmF2 is not 1/metercube but el/metercube. The unit of frequency is indicated by Hz not hz. The units are never written in italics. There should always be one blank space between the number and the unit. The asterisk is not a proper mathematical notation for multiplication.

[response] The units have been deleted from Equation (2) and the asterisk was replaced with the multiplication dot.

——————-
——————-

[referee] 11) Equation 3 does not represent the parameters of the application. For one ionosonde station that computes foF2 every 15 minutes, how can there be one value for whole month of January or July?

[response] We take a median of all observational data which fell between 10LT and 14LT for the month of January as well as July. This is described at the end of paragraph 2 of the methodology.

——————-
——————-

12) Figure3 is also very unclear. The horizontal line for 80 is missing. The main problem is that the plot is prepared for the 12 month running mean of IGNS not IGNS itself. Which input data file did the authors use in this study? The years chosen for various levels of solar activity are given in Table 3. According to these information, the study covers the years from 1970 to 2014, yet in the rest of the paper, the results are provided for 1970 to 1990 (such as Figure 4) or the years are not mentioned at all. For a list of 8 years for low activity years, there is only one value in the tables. What happened to the data? The criterion for less than 10 percent is not clear.

We will amend with a figure that includes the 80 line.
We used the 12-month running mean of the IGNS to define solar activity levels instead of a solar index such as F10.7 or sunspot number. The reasons for this are given in the penultimate paragraph of the methodology section.
The study covers observational data which span from 1970 to 2014. Figure 4, is a direct comparison with work presented by Rishbeth and Muller-Wodarg (2006) which describes the AI variation from 1970 to 1990. We will add the following text to the first paragraph of section 3.1.1

(page 12 line 1): "Data are presented for the Wallops–Hobart (green), Wakkanai–Port Stanley (blue), and Kodaikanal–Huancayo (red) station pairs for the years 1970-1990, a subset of our full data record. This subset is presented in order to recreate Figures 3 and 4 from Risbeth and Muller-Wodarg (2006) and perform a direct comparison with their work.

———————-

———————-

[referee] 13) In Tables 4, 5 and 6, there are two stations and one 'Iono' value. How is this possible? How did the authors compute the values? The caption of the titles mention that the values are AI, yet the column titles indicate that they are IRI(IG) or Iono. The 'average' and 'average*' operations are not clear. Why did the authors include station pairs with no data into the tables?

[response] Ionosonde observations are not always available, and it is important to communicate this as well to support the integrity and validity of our results.

We will include the following text in the methodology section, page 8 line 13:

(page 8, line 13) The SfoF2 and the AI are computed for various levels of solar activity. Both parameters will be computed for every year of data available the stations listed in Figure 2 and Table 2. We then group these computed values by solar activity level and average. This presents the average AI and solstitial variation for a given station at various solar cycle levels. We use the index IGNS as a solar proxy.

———————-

———————-

[referee] 14) In Table 6, the conclusions drawn are wrong. For 7 station pairs that have data, IG input matched 4 of these, whereas IGNS matched only 2. So IG is a better input for high solar activity years.

[response] Our conclusions are based on the overall average AI predicted by IRI, which

indicate IGNS is the better index. We will clarify this, with an additional error statistic.
—————-
—————-

[referee] 15) Figure 4 does not include years from 1990 to 2014. It is not clear why all the stations are not given? What is the meaning of diamonds? Some lines have them and some does not. If they are the years of computation, this information does not match Table 3. The subplots are too crowded.

[response] We address the first part of this comment in 12).
In a subsequent submission will include a legend which indicates the dotted lines correspond with ionosonde observations.

—————-
—————-

[referee] 16) The information on the computation of AI on page 12 is not clear at all. All mathematical computations should be clearly indicated with proper notation and equation numbers.

[response] We will all add a regressive equation to correspond with our explanation for figure 12.

—————-
—————-

[referee] 17) I have reservations for the 'missing data replacement' as done in the manuscript.

[response] We agree with the reviewer's reservations. Unfortunately, the choice to combine dataset was driven by a need to present a full picture, balanced by an availability of data.

—————-

——————-

[referee] 18) Table 7 does not make sense at all. The correlation coefficients are computed for which data sets? What is NPTS mean? Is the correlation biased or unbiased? For what years and for which solar activity level?

[response] Table 7 computes an unbiased correlation between all of the AI from observations and from IRI for the entire data record using each station pair.

We will add this detail to the text on page 13 line 22. "We compute the correlation coefficient between all AI values from observations and the AI values from the IRI predictions for each station pair. We use the entire data record, instead of just 1970-1990 as presented in figure 4. These results are presented in table 7. "NPTS" indicates the number of data points used for the correlation coefficient. "

——————-
——————-

[referee] 19) Figure 5 is also another mystery. Even the labels are misspelt. What does the vertical bars or lines represent? And so on...

[response] We have corrected the misspelling in table 5.
We also add the following text to 3.2 for added clarity:
(Page 14 line 14): "We draw vertical lines through data points which correspond to the same station to aid in the visual inspection of the chart".

——————-
——————-

[referee] 20) Table 8 is wrong.

[response] This comment is difficult to respond to directly unless there is a reference to contrary results from other literature. Our numbers are consistent with the observational data as well as similar analysis presented by Torr and Torr (1977) and Zhao et

al., (2008).

——————-

——————-

[referee] 21) Since the equations for computation of AI, averaging of AI and SfoF2 are totally unclear and may possibly contain significant physical and mathematical errors, none of the comments in the discussion or the drawn conclusions are reliable.

[response]

Our definition and implementation of AI is consistent with previous studies of the annual anomaly using the AI index (this is addressed in points 3-5, 20). Regarding mathematical errors, We will add the following text for added clarity:

(Page 9, line 9) "The range of the observed AI values presented in these tables are consistent with previous literature (please reference 1). "

(Page 15, line 6) These trends and numerical values of the observed SfoF2 are consistent similar analysis presented by Torr and Torr (1977).

——————-

——————-

[referee] 22) The paper is full of notational, grammatical and mathematical errors. The authors should start using at least a spell-checker and technical reviewer to edit the paper. There are syntax errors and singular-plural errors. There are many incidences of two verbs are used in the same sentence.

[response] Our document has been fully re-edited.

——————-

——————-

[referee] 23) The authors should stop 'suspecting'. 'Suspect' is not part of scientific

terminology.

[response] "We have removed 'suspect' from the body of the manuscript.

[referee] 24) There is no references for IGRF model and some of the data sources are not acknowledged.

[response] We will include the corresponding IGRF citation in the methodology (and append the reference list). We have also included an acknowledge to GIRO NDGC in our acknowledgements.

[Figure]

Figure 4.2: The 12-month running mean of $IG^{NS}$, $IG^{NS}_{12}$, as a function of time. The horizontal dotted lines indicate regions for which the solstices differences are calculated: deep low ($IG^{NS}_{12} < 8$), low ($8 < IG^{NS}_{12} < 20$), low moderate ($50 < IG^{NS}_{12} < 80$), high moderate ($100 < IG^{NS}_{12} < 135$) and high ($135 < IG^{NS}_{12}$). Only years in which $IG^{NS}_{12}$ changes by less than 10% are considered (red diamonds).

**Fig. 1.** revised figure 4.2